# Differences in workplace violence and health variables among professionals in a hospital emergency department: A descriptive-comparative study

Andrea Cascales-Martínez[1], Paloma López-Ros[2]*, David Pina[3], Juan Manuel Cánovas-Pallares[4], Reyes López López[1], Esteban Puente-López[5], Carlos Piserra Bolaños[6]

1 University of Murcia, Murcia, Spain, 2 Department of Behavioral Sciences and Health, Miguel Hernández University, Elche, Spain, 3 Department of Educational Sciences, University of La Rioja, La Rioja, Spain, 4 Nurse in the Healthcare Emergency at Comunidad Valenciana, Valenciana, Spain, 5 Department of Psychology, University of Valladolid, Valladolid, Spain, 6 Occupational Risk Prevention Service, Government of La Rioja, La Rioja, Spain

* plopez@umh.es

**Data Availability Statement:** All relevant data are within the manuscript and its Supporting Information files.

## Abstract

### Introduction

Workplace violence is a relevant social problem due to its high prevalence and serious consequences. A quarter of workplace violence occurs in the healthcare sector. Evidence shows differences among professionals, with emergency department workers being especially vulnerable, presenting a higher risk of suffering mental and physical health problems, as well as threats to their professional and social integrity.

### Objective

To explore the frequency with which emergency department professionals are exposed to user violence and violence by their own coworkers; as well as to analyze the differences between different professionals in exposure to violence in the workplace and some of its most studied consequences such as burnout, job satisfaction, engagement, and general health.

### Methods

A descriptive comparative study was carried out with a sample of 120 emergency department workers from three hospitals in Alicante. The majority were healthcare professionals (84.2%), women (61.7%), obtaining a mean age of 41.8 years (SD = 10.8). Sociodemographic and occupational variables, user violence, violence among colleagues and superiors, general health, burnout, engagement, and job satisfaction were evaluated.

### Results

A high prevalence of both physical and non-physical user violence in the healthcare setting was observed, especially affecting nursing and administrative assistants. In addition, significant differences were identified between professionals in terms of non-physical user

**Funding:** The author(s) received no specific funding for this work.

**Competing interests:** The authors declare that they have no known competing financial interests or personal relationships that could have appeared to influence the work reported in this paper.

violence, burnout, engagement, and job satisfaction. Administrative staff suffer greater non-physical user violence, while nursing assistants show higher levels of engagement. Regarding job satisfaction, nurses report higher intrinsic satisfaction. Medical staff, nurses and nursing assistants show higher levels of extrinsic satisfaction compared to administrative staff.

## Discussion

Our results are consistent with other studies in which a relationship between exposure to violence and job satisfaction is observed. In addition, administrative staff appear to be the professionals most exposed to violence from both patients and coworkers. These results provide evidence for future research focused on improving the work environment and health of emergency department professionals.

## 1. Introduction

Workplace violence is an area of growing interest in both research and prevention/intervention. This type of violence has been defined as "any unreasonable action, incident or behavior by which a person is assaulted, threatened, humiliated or injured by another person in the course of or as a direct consequence of his or her professional activity" [1]. This definition encompasses sexual violence, hitting, kicking, pinching, slapping, injury by weapons, stabbing, biting, abuse of power, threats of physical force, verbal abuse, psychological harassment or intimidation, among other manifestations of violence [1–3]. Workplace violence is an occupational hazard with implications for the safety and health of workers, and has mobilized the main international institutions. For example, the European Commission, through the new OSH framework 2021–2027, promotes safe and healthy working conditions and ratifies the former Violence and Harassment Convention 2019 (No. 190) of the International Labor Organization by strengthening sanctions, establishing new laws and policy measures to prevent and address workplace violence and harassment [4].

In the healthcare sector, violence has become a phenomenon of great interest due to its growing increase [5–9]. According to the Occupational Safety and Health Administration's classification of workplace violence [10], healthcare personnel can be subjected to different sources of violence depending on the relationship with the perpetrator(s). The most studied type of violence in this sector is type II violence, which refers to violence perpetrated by clients, patients or consumers and their relatives. Regarding its prevalence, there seems to be an underreporting of the actual data due to the normalization of these violent acts by professionals and the lack of reporting [11, 12]. However, recent reviews provide worrying values with an overall prevalence of 61.9%, with non-physical violence being more frequent (42.5%) compared to physical violence (24.4%). Regarding specific behaviors, verbal aggression is the most common, followed by threats and sexual harassment [13].

There are also differences according to the professional category, the type of care provided to users or the professional activity. In this regard, a higher frequency is observed in hospital emergency departments and psychiatry [13, 14], reaching a rate of up to 100% of professionals in countries such as Spain [15]. Regarding the professional group, nursing and medical staff have generally been reported as the most exposed (59.2% and 56.8%) [13], although other studies have shown that non-physical violence is more frequent among administrative staff, and physical violence is more frequent among nursing assistants [15].

In addition to user or type II violence, healthcare personnel are also frequently subjected to type III workplace violence, characterized by the prior relationship between the agents involved [10]. In other words, this type includes violence between employees, whether they are superiors (vertical violence) or their own colleagues (horizontal violence). This type of violence can be personal (repetitive criticism, humiliation, spreading gossip, practical jokes, etc.), occupational (work overload, excessive control, manipulation of working conditions, etc.) or social (isolation, exclusion, etc.) [16, 17]. Recently, Vidal-Alves et al. [16] estimated the prevalence of this type of violence to be around 59%, with personal violence being most frequent (51%), followed by social violence (37.3%) and, finally, workplace violence (21.3%). In general terms, type III workplace violence has been less studied in comparison with other types of violence, although differences have also been reported according to sex, time in the profession, age, type of unit [16], professional activity [18] or type of shift [19]. Male workers with less than 10 years in the profession appear to be at greater risk of personal violence and those in outpatient services and ambulatory units social and occupational violence [16]. In addition, nursing staff, administrative staff, and health care assistants [18] along with variable shift workers have been identified as those at higher risk for peer violence [20].

Generally speaking, a work environment exposed to violence has negative effects on professionals in the short, medium and long term. In emergency departments, Hassankhani et al. [21] identified 4 major blocks affected by exposure to violence: (a) mental health, (b) physical health, (c) professional integrity, and (d) social integrity. Specifically, it has been observed that healthcare personnel who are victims of violence manifest feelings related to loss of time, post-traumatic stress disorder, depression, lack of sense of safety or lack of protection, and emotional distress including anger, humiliation, fear, and guilt [9, 22]. In addition, an impairment of job satisfaction has been reported. The greater the exposure, the more professionals feel less extrinsic and intrinsic job satisfaction, the more harmful effects on their perceived health or quality of life [8, 16, 23]. One of the most studied variables in this population is burnout syndrome [24]. In this regard, the occurrence of burnout syndrome in healthcare professionals seems to be associated both with exposure to violence and with the levels of self-esteem and/or empathy of the professionals [16, 25]. In addition, staff who are victims of violence, psychological harassment and/or threats show lower levels of engagement [26, 27], which can influence patient satisfaction, the work environment, the quality of the services provided and the frequency of errors [28, 29]. This situation in healthcare personnel can lead to increased absenteeism [9], lower quality of care and/or lower productivity [6] among other multiple consequences for the organization.

Despite the relevance of the situation, previous studies carried out in Spain in emergency departments have focused mainly on user or type II violence [15, 26, 27, 30, 31] and its relationship with burnout [8, 16, 25, 30]. Among the professional groups, nursing staff has been the most studied [16, 25–27, 32]. In other words, there are hardly any studies that explore both type II and type III violence in emergency department professionals, taking into account the different professional groups that make up this service [8, 16, 25, 27, 30]. For this reason, the present study aims to explore the perception of user violence and lateral violence in emergency department professionals. Specifically, it will delve into the differences between the main professional groups in emergency departments regarding exposure to user violence, lateral violence, burnout, job satisfaction, engagement and general health.

## 2. Materials and methods

An empirical cross-sectional study was carried out with a quantitative and descriptive-associative methodology in the emergency department of three public hospitals located in southeastern Spain. Data were collected between 1 October and 1 December 2021.

## 2.1. Participants

The final sample size consisted of 120 healthcare professionals (84.2%) and non-healthcare professionals (15.8%) working in hospital emergency departments. According to the characteristics of the sample (Table 1), the majority were medical (47.5%), nursing (23.3%), nursing assistants (13.3%), administrative (9.2%) and other professions (e.g. orderlies) (6.7%). The majority were women (61.7%) of Spanish nationality (90.8%) and married (49.2%). The mean age was 41.8 years (SD = 10.8). Of the workers, 50.8% had a permanent contract with a length of service of 6.9 years and a length of service in the profession of 13 years. In addition, the frequency of the work shift was mostly 12 hours (60.2%).

## 2.2 Instruments

A questionnaire with a protocol of 116 items was administered. It included the collection of sociodemographic and occupational data by means of an *ad hoc* questionnaire (Table 1). Other scales measuring user violence, violence by colleagues and superiors, general health, burnout, engagement, and job satisfaction were also used.

**2.2.1 Aggressive behavior scale for health care workers users (HABS-U) [33].** This instrument evaluates the perception of the professionals on the aggressive behaviors of the users towards the health personnel in the last year. It consists of 10 items with 6 response options (1 = never; 6 = daily). In the original article, a two-factor factor structure was found: non-physical violence (seven items) with a Cronbach's alpha of .84, and physical violence (three items) with a Cronbach's alpha of 0.76. In our sample, adequate reliability values were obtained for the non-physical violence factor (α = .902; ω = .905). However, for the physical violence factor reliability was lower (α = .590; ω = .698).

**2.2.2 Aggressive behavior scale hospital: Coworkers and superiors (HABS-CS) [33].** This scale assesses the frequency in the last year of low and medium intensity violence between colleagues or from superiors to employees in the healthcare context. This questionnaire consists of 17 items grouped into five factors: active-superior workplace bullying (4 items), personal lateral violence (4 items), relational lateral violence (3 items), passive-superior workplace bullying (3 items) and lateral workplace violence (3 items). Other studies find adequate general reliability of this instrument (α = .863). In our study, the scales of active and passive-superior workplace violence showed excessively low reliability values and it was therefore decided to exclude them. As for the factors related to peer violence, moderate reliability was obtained, with McDonald's omegas of .691, .684, .689, for personal, relational, and workplace lateral violence, respectively.

**2.2.3 Goldberg General Health Questionnaire version 28 items (GHQ-28).** An instrument originally developed by Golberg and Hillier (1979) [34]. This questionnaire assesses general health and detects problems of social dysfunction, psychosomatics, anxiety and depression. The Spanish adaptation of Lobo et al. (1986) is composed of 28 items in a 4-category response format (1 = not at all; 4 = much more than usual), grouped into 4 subscales: psychological somatic symptoms (somatic GHQ-7 items), anxiety and insomnia (anxiety GHQ-7 items), social dysfunction scale (dysfunction GHQ-7 items) and depressive symptoms scale (depression GHQ-7 items) [35]. Regarding the reliability of each subscale in our sample, adequate McDonald's omega values were obtained for the somatic symptoms (ω = .800), Anxiety (ω = .905) and social dysfunction (ω = .749) factors. However, for the depression factor, a medium-low value was obtained (ω = .612).

**2.2.4 Maslach Burnout Inventory-General Survey (MBI-GS).** The MBI-GS questionnaire, originally developed by Schaufeli et al. (1996) [36] and adapted, translated, and validated in Spanish by Gil-Monte et al. (2002) [37] was used. This questionnaire evaluates the degree of

**Table 1. Sociodemographic and labor characteristics of the sample.**

|  | *n* | % |
|---|---|---|
| Age (years) |  |  |
| 35 or less | 41 | 34.2% |
| 36–45 | 36 | 30.0% |
| 46–55 | 26 | 21.7% |
| 56–65 | 16 | 13.3% |
| Missing data | 1 | 0.8% |
| TOTAL | 120 | 100% |
| Sex |  |  |
| Male | 46 | 38.3% |
| Female | 74 | 61.7% |
| TOTAL | 120 | 100% |
| Nationality |  |  |
| Spanish | 108 | 90.8% |
| European | 4 | 3.4% |
| Non-european | 6 | 5.0% |
| Missing data | 2 | 0.8% |
| TOTAL | 120 | 100% |
| Civil status |  |  |
| Married or unmarried couple | 59 | 49.1% |
| Single | 53 | 44.2% |
| Divorced or widowed | 8 | 6.7% |
| TOTAL | 120 | 100.% |
| Professional group |  |  |
| Medical staff | 57 | 47.5% |
| Nursing staff | 28 | 23.3% |
| Auxiliary nursing staff | 16 | 13.3% |
| Non-medical personnel | 11 | 9.2% |
| Others | 8 | 6.7% |
| TOTAL | 120 | 100% |
| Speciality |  |  |
| General Emergency | 107 | 89.1% |
| Pediatric Emergency | 7 | 5.8% |
| Mental health Emergency | 2 | 1.5% |
| Other | 2 | 1.5% |
| Missing data | 2 | 1.5% |
| TOTAL | 120 | 100% |
| Type of contract |  |  |
| Permanent contract | 61 | 50.8% |
| Temporary contract | 51 | 42.5% |
| Open-ended contract | 8 | 6.7% |
| TOTAL | 120 | 100% |
| Seniority (years) |  |  |
| 0–2 | 26 | 21.7% |
| 3–5 | 24 | 20% |
| 6–10 | 19 | 15.8% |
| 11–15 | 38 | 31.7% |
| Missing data | 13 | 10.8% |

(*Continued*)

**Table 1.** (Continued)

| | *n* | % |
|---|---|---|
| TOTAL | 120 | 100% |
| Experience in the profession (years) | | |
| 0–10 | 44 | 36.6% |
| 11–20 | 46 | 38.3% |
| 21–30 | 11 | 9.2% |
| +30 | 6 | 5% |
| Missing data | 13 | 10.9% |
| TOTAL | 120 | 100% |
| Work shift | | |
| 12 hours | 68 | 56.7% |
| 7 hours | 18 | 15% |
| Others | 27 | 22.5% |
| Missing data | 7 | 5,8% |
| TOTAL | 120 | 100% |
| Ongoing training | | |
| Yes | 71 | 59.2% |
| No | 30 | 25% |
| Missing data | 19 | 15.8% |
| TOTAL | 120 | 100% |
| Sick leave in the last 12 months | | |
| No | 70 | 72.2% |
| Yes | 20 | 20.6% |
| Missing data | 30 | 7.2% |
| TOTAL | 120 | 100% |

Burnout of workers. It consists of 15 items distributed in three dimensions: emotional exhaustion (5 items), professional efficacy (6 items) and cynicism (4 items). The response format ranges from 1 (never/never) to 7 (always/every day). Other studies find good internal consistency, with Cronbach's alpha of .83 for emotional exhaustion, .85 for professional efficacy and .74 for cynicism. In our study, adequate reliability values were obtained for each of the scales: emotional exhaustion ($\alpha$ = .831; $\omega$ = .841), professional efficacy ($\alpha$ = .836; $\omega$ = .841) and cynicism ($\alpha$ = .680; $\omega$ = .718).

**2.2.5 Utrecht Work Engagement Scale (UWES-9).** There are different versions of the Utrecht Work Engagement Scale, with 17, 15 and 9 items. All versions evaluate engagement, understanding that this includes the variables of vigor, dedication, and absorption at work. For this study, the short version of 9 items translated into Spanish [38] was used. These 9 items are distributed in 3 subscales of 3 items each: vigor, dedication, and absorption. The items are answered on a Likert scale from 1 (never) to 7 (always). The internal consistency assessment shows Cronbach's $\alpha$ values of .84 for the vigor subscale, .89 for dedication and .79 for absorption. In our sample, adequate reliability values are also observed: vigor ($\alpha$ = .601; $\omega$ = .734), dedication ($\alpha$ = .861; $\omega$ = .866) and absorption ($\alpha$ = .797; $\omega$ = .800).

**2.2.6 Minnesota Satisfaction Questionnaire (MSQ) [39].** This instrument evaluates job satisfaction; there are two versions, a long version of 100 items and a short version of 20. In the present study, the short version translated into Spanish and adapted by Gricélidys-Rodríguez [40] was used. This questionnaire has a Likert-type scale with 5 response options ranging from very dissatisfied to very satisfied. The original study considers that the scale is made up of

three subscales: One of intrinsic satisfaction (12 items), another of extrinsic satisfaction (6 items), and general satisfaction (2 items). As for the reliability observed in our sample, the McDonald's Omega was .835 for intrinsic satisfaction and .756 for extrinsic satisfaction. In addition, the overall scale presented a McDonald's Omega of .900.

## 2.3. Procedure

The present study was approved by the Research Ethics Committee of the authors' university (AUT.DCC.PLR.231123). Participation was voluntary and anonymous, following the guidelines of the Data Protection Law. Approval was also obtained from the different management and/or executive teams of the participating centers. All study participants signed an informed consent form.

As a recruitment measure, we first invited 6 public hospital emergency centers with the collaboration of the HURGE project (Humanization of Emergency and Urgent Care) located in a community in southeastern Spain.

3 of the centers that were invited showed interest in participating on the research: Hospital Universitario de Virgen de los Lirios, Hospital Universitario de Torrevieja y Hospital Universitario del Vinalopó. The total number of workers in emergency services corresponds to 328. All centers have a similar number of health professionals on staff in all specialties, with a total of 73 medical professionals (between 23–25 per center), 76 auxiliary nursing professionals (between 23–28 per center) and 123 nursing professionals (between 37–44 per center)

A detailed report of the study, its development and confidentiality form was previously given to the corresponding management teams. Subsequently, a person responsible for each center was assigned, informed, and trained in the application of the evaluation protocol. The sample was selected by random cluster sampling with healthcare and non-healthcare personnel from the hospital emergency departments of the participating centers. The questionnaire was administered randomly in paper and pencil format to 50% of the staff in each unit, with the aim of adjusting to the sample size recommendation obtained in the statistical power analysis. It was ensured that all professional categories had the same percentage of participation. A response rate of 73.17% was obtained.

The questionnaires were collected in sealed envelopes one month after administration. These envelopes were not labeled, and the responses were sent via email to the research teams.

## 2.4. Data analysis

The data were analyzed using the JAMOVI program [41]. Statistical power analyses were performed using G-Power [42] to estimate the minimum sample size necessary for the results to be sufficiently consistent. Given the effects observed in previous similar studies conducted in the context, a minimum effect size of 0.3 was used, with a minimum desired power of 0.9 and an α of 0.05, yielding a required sample size of 119 participants.

Once the data had been collected, descriptive and frequency analyses were initially performed to determine the characteristics of our sample. In addition, the mean and standard deviation of the instruments used were calculated, and the reliability of the questionnaires was checked by calculating Cronbach's alpha and McDonald's omega. On the other hand, the scores of the variables studied were calculated through the mean of the items that formed them. The specific items are shown in Supplementary File 1.

Subsequently, the prevalence of exposure to violence among users and among professionals was calculated. For this, it was considered that, when the professional responded "2 = Annually", "3 = Quarterly", "4 = Monthly", "5 = Weekly" o "6 = Daily"; they had experienced some

form of violence in the workplace at least once in the past year. Subsequently, the percentages of perceived workplace violence by professional group and year were calculated.

The Shapiro-Wilk and Levene tests were used to analyze normality and homogeneity, respectively. The p levels were significantly low ($p < .05$), not complying with these assumptions. The literature recommends the use of nonparametric statistics when the assumptions of normality and homogeneity are not met [43]. Therefore, a Kruskal-Wallis test was performed to test for differences between the different professionals in the variables described above. Effect sizes ($ε2$) were also calculated, considering for its interpretation Cohen's criteria for the partial eta2 analogous test: 0.01 (small), 0.06 (moderate) and 0.14 (large). This is recommended by previous literature [44]. Finally, Dwass-Steel-Critchlow-Critchlow-Fligner pairwise comparisons were performed.

## 3. Results

A high prevalence of non-physical violence by users was observed. 100% of the auxiliary nursing staff, administrative assistants and other professionals, 96.42% of the nursing staff and 90.09% of the medical staff have experienced an episode of such violence at least once in the last year (Table 2). Most frequently, users get angry because of delays in care.

Regarding physical violence by users, 50% of the nursing assistants have suffered some episode in the last year, followed by 45.45% of the administrative staff, 25.45% of the medical staff and 25% of the nursing staff and other professionals (Table 2). In this regard, it is most common for users to show their anger at the destruction of material or infrastructures.

On the other hand, 68.75% of the nursing assistants have experienced some episode of personal lateral violence in the last year, followed by 62.5% of other professionals, 47.27% of the medical staff, 46.42% of the nursing staff and 45.45% of the administrative staff, the most frequent being ironic jokes by colleagues. Lateral relational violence, although less frequent, is manifested in 31.25% of the nursing assistants, followed by 18.18% of the administrative staff, 14.54% of the medical staff, 14.2% of the nursing staff and 12.5% of other professionals. The most frequent in this regard is that colleagues stop talking or make bad faces. Finally, lateral workplace violence is most frequent in nursing assistants (31.25%), followed by administrative staff (27.27%), nursing staff (17.85%), medical staff (16.36%) and no one from the category "other professionals" (Table 2).

Table 3 shows the results of the Kruskal-Wallis test, the effect size and the post hoc comparisons between the different professionals on the variables studied. Significant differences are observed in the non-physical user violence variables ($χ^2 = 13.36$; $p < .05$), with a moderate effect size of 0.11. Post hoc analyses revealed that these differences are significant between medical staff and administrators, the latter being those who suffer greater non-physical violence by users.

**Table 2.  Percentage of perceived workplace violence by occupational group per year.**

| Type of violence | Medical staff | Nursing staff | Auxiliary nursing staff | Non-medical personnel | Other |
|---|---|---|---|---|---|
| User violence | | | | | |
| Non-physical | 90.09 | 96.42 | 100 | 100 | 100 |
| Physical | 25.45 | 25 | 50 | 45.45 | 25 |
| Lateral violence | | | | | |
| Personal | 47.27 | 46.42 | 68.75 | 45.45 | 62.5 |
| Relational | 14.54 | 14.2 | 31.25 | 18.18 | 12.5 |
| Workplace | 16.36 | 17.8 | 31.25 | 27.27 | 0 |

**Table 3. Differences among professionals in hospital emergency services in variables of violence, health, engagement and satisfaction.**

| Variable | | Medical staff (A) | Nursing staff (B) | Auxiliary nursing staff (C) | Non-medical personnel (D) | Other (E) | Kruskal-Wallis | | |
|---|---|---|---|---|---|---|---|---|---|
| | | M (SD) | M (SD) | M (SD) | M (SD) | M (SD) | $\chi^2$ | $\varepsilon^2$ | Post hoc |
| User violence | | | | | | | | | |
| | Physical | 1.27 (0.54) | 1.30 (0.45) | 1.67 (0.93) | 1.61 (0.63) | 1.42 (0.79) | 6.99 | 0.06 | - |
| | Non-physical | 3.08 (1.27) | 3.36 (1.19) | 2.96 (1.45) | 4.56 (1.20) | 3.92 (1.55) | 13.36** | 0.11 | A-D |
| Lateral violence | | | | | | | | | |
| | Personal | 1.50 (0.62) | 1.48 (0.65) | 1.90 (0.85) | 1.50 (0.67) | 1.72 (0.63) | 6.01 | 0.05 | - |
| | Relational | 1.14 (0.24) | 1.15 (0.38) | 1.46 (0.74) | 1.27 (0.69) | 1.13 (0.35) | 5.60 | 0.04 | - |
| | Workplace | 1.17 (0.42) | 1.19 (0.38) | 1.49 (0.93) | 1.24 (0.37) | 1.00 (0.00) | 6.07 | 0.05 | - |
| General Health | | | | | | | | | |
| | Depression | 1.23 (0.29) | 1.13 (0.22) | 1.13 (0.16) | 1.25 (0.20) | 1.09 (0.17) | 5.97 | 0.05 | - |
| | Anxiety | 2.11 (0.83) | 2.01 (0.67) | 2.01 (0.70) | 1.94 (0.70) | 2.11 (0.75) | 0.49 | 0.004 | - |
| | Somatic symptoms | 2.04 (0.73) | 1.96 (0.52) | 2.00 (0.66) | 1.95 (0.80) | 2.18 (0.85) | 1.00 | 0.008 | - |
| | Social dysfunction | 1.42 (0.50) | 1.32 (0.56) | 1.36 (0.38) | 1.61 (0.48) | 0.92 (0.57) | 8.02 | 0.06 | - |
| Burnout | | | | | | | | | |
| | Emotional exhaustion | 3.56 (1.42) | 3.16 (1.12) | 2.57 (1.24) | 3.51 (1.34) | 2.33 (1.08) | 9.54* | 0.08 | - |
| | Professional efficiency | 5.07 (1.17) | 5.25 (1.03) | 5.10 (1.43) | 4.76 (1.40) | 6.33 (0.85) | 10.30* | 0.08 | AB-E |
| | Cynicism | 2.69 (1.34) | 2.33 (1.01) | 2.05 (1.21) | 3.07 (1.39) | 1.91 (0.95) | 6.99 | 0.06 | - |
| Engagement | | | | | | | | | |
| | Vigor | 4.40 (1.77) | 5.14 (1.26) | 5.58 (0.99) | 4.85 (1.60) | 5.50 (1.32) | 14.67** | 0.12 | A-C |
| | Dedication | 4.65 (1.27) | 5.45 (1.17) | 5.94 (0.79) | 4.67 (1.51) | 5.67 (1.43) | 17.25** | 0.14 | A-C |
| | Absorption | 4.07 (1.44) | 5.05 (1.39) | 5.31 (0.92) | 3.72 (1.14) | 4.96 (1.35) | 16.81** | 0.15 | A-BC |
| Job satisfaction | | | | | | | | | |
| | Intrinsic | 3.81 (0.46) | 4.17 (0.53) | 3.94 (0.26) | 3.31 (0.05) | 3.68 (0.60) | 12.98* | 0.14 | B-D |
| | Extrinsic | 3.28 (0.75) | 4.02 (0.49) | 3.89 (0.40) | 2.98 (0.61) | 3.60 (0.58) | 30.21*** | 0.27 | A-BC-D |
| | General | 3.55 (0.84) | 4.00 (0.68) | 3.88 (0.50) | 3.30 (0.82) | 3.81 (0.80) | 10.37* | 0.08 | - |

Notes: p<0.05*; p<0.01**; p<0.001

Significant differences were also observed in burnout, specifically in emotional exhaustion ($\chi^2$ = 9.54; $p<$ .05) and professional efficacy ($\chi^2$ = 10.30; $p<$ .05) with moderate effect sizes. However, post hoc analyses show differences only in professional efficacy between medical staff and nurses, with respect to other professionals.

For the factors related to engagement, significant differences were observed in vigor ($\chi^2$ = 14.67; $p<$ .05) and dedication ($\chi^2$ = 17.25; $p<$ .05), between medical staff and nursing assistants, with moderate effect sizes. On the other hand, in absorption ($\chi^2$ = 16.81; $p<$ .05), differences were found between medical staff, with respect to nurses and assistants; also with a moderate effect size. In all cases, the mean of the nursing assistants group showed higher scores in engagement ($\chi^2$ = 16.81; $p<$ .05).

Finally, in job satisfaction, significant differences were obtained in intrinsic satisfaction ($\chi^2$ = 12.98; $p<$ .05), with a moderate effect size, being higher in the nursing professional group compared to the administration group. In extrinsic satisfaction, medical staff, nursing staff and nursing assistants showed significantly higher scores than the administration group ($\chi^2$ = 30.21; p< .001). In this case, the magnitude of the effect was high.

## 4. Discussion

The aim of this study was to analyze the differences that exist among professionals in a hospital emergency department in terms of user violence (type II) and violence among professionals (type III), as well as in some of their most relevant consequences. To date, there are hardly any studies that explore this relationship in this population, so it is necessary to expand the available evidence in order to understand how exposed the different professionals in hospital emergency departments are to workplace violence; and how this exposure affects each worker. Several studies have pointed out the need to specifically evaluate the different contexts with the aim of designing and implementing prevention and intervention programs in order to improve the occupational health of healthcare professionals [45, 46].

Thus, significant differences are observed among professionals in the frequency and way in which they are exposed to non-physical violence at work by the user. Moreover, these differences are also identified in variables such as emotional exhaustion, professional effectiveness, engagement, and job satisfaction. In contrast, among our results we found no significant differences in user physical violence, peer violence, cynicism, and general health. An important finding is that administrative staff are the most exposed in relation to non-physical violence by users. This has been previously reported both in emergency departments [15, 30] and in Primary Care [47]. The main triggers of user violence are related to administrative and supply aspects [48]. Administrative staff, being the first instance of care, are highly exposed to verbal aggressions, as users often unload their anxiety and fear on them, in addition to being a professional group that deals with a greater number of users [49]. However, there seems to be no consensus in the literature as to which healthcare group is more vulnerable to any type of aggression. A review by Liu et al. [13] concluded that at the international level, nurses are the most exposed to all forms of violence. These differences between the results obtained in the Spanish context as opposed to the international literature could be explained by the characteristics of the healthcare system itself. In both Emergency and Primary Care in Spain, the administration staff is the patient's first contact. In this line, it has been pointed out that when the nursing staff is in charge of this first contact, the risk of suffering violence by users in this professional group is higher than in others [50]. Within non-physical violence, verbal aggression seems to be the most frequent among emergency personnel, including threats, intimidation, and insults [13, 51, 52]. Delays in care, lack of information and lack of personnel seem to be the main reasons for user anger [45, 51, 53].

In our study, no differences were observed between the different professionals in terms of lateral violence and, in general terms, this seems to be less frequent than user violence. Previous literature has focused mainly on nursing staff, delving into the consequences of this phenomenon, such as burnout, poor psychological indicators, considering quitting their job or the impact on patient care and attention [16, 54]. The results of our study are consistent with previous studies. The literature indicates that profiles of nurses affected by lateral violence often show a significant negative correlation with both intrinsic and extrinsic job satisfaction, and a positive correlation with emotional exhaustion, somatic symptoms, anxiety, insomnia, social dysfunction, and depression [25, 32]. Although this field has been widely studied in nursing professionals, no studies have been found showing possible differences between professional groups in emergency departments. Although exposure to this violence is relatively low, its study is essential to make visible the severe consequences that this type of situation can have on emergency department professionals [16, 25].

Some of the variables associated with lateral violence are hospital management, personal characteristics of perpetrators and victims, such as age and gender [55]. Regarding gender, significant differences were found between sexes, being more common in men [25]. Other studies highlight normalization of this type of violence could be due to the perception of these actions as "normal in environments with too many women" [56]. There does not seem to be a clear consensus regarding the most vulnerable age for this type of violence. On the one hand, some studies indicate that younger staff are the most vulnerable [57], but on the other hand, other studies point to older ages and seniority as a risk factor [16].

Emergency departments have recently been considered one of the most vulnerable to workplace violence and mental health problems [58, 59]. Most emergency department staff are unaware of or consider existing resources to be scarce or unhelpful, especially in conflict situations [53, 60]. It is estimated that productivity decreases by 50% between 6 and 18 weeks after the incident, and consequently, costs for the organization also increase by 30–40% [58]. Evidence suggests a high prevalence of mental health problems in healthcare professionals [61]. Specifically, Matthews et al. [62], concluded that there is a high prevalence of Post Traumatic Stress Disorder (PTSD), depression and anxiety in hospital emergency personnel; pointing out that such symptomatology is related to the workplace itself and maladaptive coping.

Although in our study no significant differences were observed in the general health of emergency professionals, the scores found show high levels of anxious and somatic symptoms. In previous studies, exposure to workplace violence and this type of symptomatology have been directly related [25, 30, 63], including in other services, such as primary care [12]. These symptoms appear to be more frequent in staff with fewer years in the profession, which is attributed to limited experience, patient overload, time limits and intense working conditions [64].

Regarding burnout, medical personnel present higher levels of emotional exhaustion compared to other professionals. This syndrome has been widely studied among emergency personnel, due to its high frequency in this sector [64, 65] and because of its relationship with workplace violence [8, 66]. Durand et al. [65], identified the high level of psychological demands of medical personnel, identifying it as a predisposing factor to high levels of emotional exhaustion. Thus, sleep disturbances caused by night work, present in emergency departments, have also been associated with burnout. This type of symptomatology is not exclusive to medical personnel. Although to a lesser extent, nursing staff also showed differences with other professionals in burnout, showing lower professional efficacy. Shift work, low job satisfaction and low quality of work life seem to be the factors most associated with burnout in this professional group [67]. Regarding cynicism, no significant differences were observed in our sample. However, recent studies indicate that working conditions in the

emergency department may be related to low mental health status, increased motivation to leave the department, and a higher level of cynicism [65].

Our results also show that nursing assistants have higher engagement (vigor, dedication, and absorption at work) compared to medical staff. Fukuzaki et al. [68], state that age, gender, job demands, resources and positive affectivity are positively associated with engagement, while negative affectivity is inversely related. In this regard, it has been observed that professionals in patient care roles with high adaptive capacity, spirituality, social support [69], professional self-efficacy and life satisfaction [66] were more likely to be engaged. In contrast, professionals with greater anxious-depressive symptomatology [69], high levels of burnout [70] stress [71] and who have been exposed to situations of workplace violence are less likely to be engaged [26, 27].

In terms of job satisfaction, nursing professionals are those with the highest indicators, as opposed to administrative staff, in line with previous studies [67]. Again, it appears that age and experience seems to be an influential factor in job satisfaction; nursing professionals aged 40 years or older report loving their work, being satisfied with the service and having a good quality of life [67]. The literature has frequently associated job satisfaction with rates of both user and interprofessional violence. Xiao et al. (2024) [72], in a recent study of nursing and medical personnel, affirms that only 11.2% of personnel who have suffered health care workplace violence continue to trust their profession and only 2% support their children to study careers related to the health sector, presenting low levels of satisfaction. In turn, job satisfaction is also affected in this sense by exposure to lateral violence in healthcare personnel [32].

These findings can contribute to the creation, or improvement, of intervention programs aimed at healthcare professionals addressing the aspects detected. The literature focuses on interventions aimed at teaching professionals about harassment in healthcare and training them, increasing their knowledge about different types of communication [73]. Based on our findings, it would be of great interest to propose programs that promote the three components of engagement (vigor, dedication and commitment). These variables protect workers from suffering higher levels of work stress; professionals who experience a high level of engagement feel more satisfied with their work [74,75]. Considering the findings of this study, special emphasis should be placed on the medical staff. At the same time, we propose specific programs aimed at administrative staff with guidelines for emotional containment in order to avoid escalation of conflicts that contribute to an increase in aggressions [46]. Finally, we propose to offer spaces of relief, which help the emotional management of situations of violence [45].

Therefore, future research could focus on evaluating the effectiveness of preventive interventions tailored to each professional group, with special focus on administrative staff, given their higher exposure to non-physical violence. These interventions could be focused on reducing violence while simultaneously increasing job satisfaction and commitment. It would also be relevant to explore the impact of both lateral and user violence on other psychological variables not assessed in this study, such as post-traumatic stress, resilience, or emotional management. Similarly, it would be of interest to explore risk-related sociodemographic variables such as gender, years in the profession, and type of contract, and how these may be linked to possible consequences. In the long term, the literature could examine the effects of exposure to violence on the physical and mental health of emergency professionals, providing a more comprehensive understanding of the consequences of this phenomenon.

## 5. Study limitations

Finally, the results should be interpreted with caution due to the limitations of our study. First, we used a descriptive-comparative cross-sectional design, which does not allow us to analyze

causal relationships. Although the sample size adjusts to the calculated one, it is limited and localized in one region. In this sense, studies with a larger sample size and from different contexts would make it possible to obtain results with greater external validity. On the other hand, the internal consistency of some of the scales used is somewhat low in our sample, which could affect the reliability of the results obtained; therefore, some variables were eliminated. Finally, further studies along these lines are needed, incorporating the analysis according to sex, age and other relevant variables not taken into account in the present study [58, 59].

## Supporting information

**S1 File. Items belonging to each variable.**
(DOCX)

## Acknowledgments

We would like to thank the professionals who participated in this study and the Valencian Health Service for their collaboration and aid to the efforts of the authors during research.

## Author Contributions

**Conceptualization:** David Pina, Juan Manuel Cánovas-Pallares, Reyes López López, Esteban Puente-López.

**Data curation:** Juan Manuel Cánovas-Pallares.

**Formal analysis:** Andrea Cascales-Martínez, Paloma López-Ros.

**Funding acquisition:** Paloma López-Ros.

**Methodology:** Andrea Cascales-Martínez, Paloma López-Ros, David Pina.

**Project administration:** Andrea Cascales-Martínez, Paloma López-Ros.

**Resources:** Juan Manuel Cánovas-Pallares.

**Supervision:** David Pina, Esteban Puente-López, Carlos Piserra Bolaños.

**Writing – original draft:** Andrea Cascales-Martínez, Paloma López-Ros.

**Writing – review & editing:** David Pina, Juan Manuel Cánovas-Pallares, Reyes López López, Esteban Puente-López, Carlos Piserra Bolaños.

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
