## [Decision Letter · Decision Letter 0]

25 Sep 2024

PONE-D-24-23201DIFFERENCES IN WORKPLACE VIOLENCE AND HEALTH VARIABLES AMONG PROFESSIONALS IN A HOSPITAL EMERGENCY DEPARTMENT: A DESCRIPTIVE-COMPARATIVE STUDYPLOS ONE

Dear Dr. López-Ros,

Thank you for submitting your manuscript to PLOS ONE. After careful consideration, we feel that it has merit but does not fully meet PLOS ONE’s publication criteria as it currently stands. Therefore, we invite you to submit a revised version of the manuscript that addresses the points raised during the review process.

We look forward to receiving your revised manuscript.

Kind regards,

Alejandro Botero Carvajal, MD

Academic Editor

PLOS ONE

2. Please include your tables as part of your main manuscript and remove the individual files. Please note that supplementary tables should be uploaded as separate "supporting information" files.

Reviewers' comments:

Reviewer's Responses to Questions

**Comments to the Author**

1. Is the manuscript technically sound, and do the data support the conclusions?

Reviewer #1: Yes

Reviewer #2: Partly

2. Has the statistical analysis been performed appropriately and rigorously? 

Reviewer #1: Yes

Reviewer #2: No

3. Have the authors made all data underlying the findings in their manuscript fully available?

Reviewer #1: Yes

Reviewer #2: Yes

4. Is the manuscript presented in an intelligible fashion and written in standard English?

Reviewer #1: Yes

Reviewer #2: Yes

5. Review Comments to the Author

Reviewer #1: Thank you for the opportunity to review this manuscript. The topic is very interesting, and the article is well-written with a clearly defined study objective. However, I have a few suggestions and questions that could enhance the manuscript:

Sample Selection: The paper should provide a more detailed account of how the sample was selected. Specifically, could you elaborate on the criteria and process used for sample selection?

Data Analysis: The data analysis section mentions the use of non-parametric tests due to violations of normality and homogeneity assumptions. Could you provide more details on how these assumptions were tested before deciding to use non-parametric methods? Additionally, how were effect sizes interpreted, and what criteria were used to determine their relevance within the context of the study?

Results Discussion: The results should be discussed in greater depth. Particularly, it would be helpful to better justify why administrative staff seem to be more exposed to non-physical violence. How do these findings compare with existing literature on this topic?

Lateral Violence: Regarding lateral violence, how do your findings align with the existing literature on this type of violence in healthcare settings? What specific factors might be contributing to the observed differences in prevalence among the various professional groups?

Practical Recommendations: While the discussion covers the implications of the results well, it would benefit from a more detailed exploration of practical recommendations for managing workplace violence based on your findings. What specific intervention programs would you propose for administrative staff and healthcare professionals?

Future Research: The discussion would benefit from a more comprehensive reflection on future research directions. What additional studies do you suggest to further explore the issues identified in your research?

Thank you again for the opportunity to review your manuscript.

Reviewer #2: The manuscript is well written, but there are some questions to review in the Methodology and Results sections:

- The authors do not specify clearly the procedure for the recruitement of the participants. They select 50 % of the staff of the Emergency departments involved but it is not clear if they are randomly selected or not. It seems that the response rate is 100 %, which seems rare in a questionaire with 116 items.

- There is too much confussion regarding the profesional profile of the participants: sometimes the authors refer to Non-medical staff and sometimes to Administrative assistants or Adminsitrators. And the “Other professionals” category sometimes is included in the analysis and sometimes not.

- The authors do not specify how they manage the results of the questionaires to analyze them: It seems that they use a collapsed measure of the different factors, but they not specify how these measures are constructed from the original ítems. Besides, they use in Table 2 a YES/NO category for user and lateral violences, but in Table 3 they use a seemingly Likert scale for the same variable.

- There is no bibliographical reference for the JAMOVI program used in the analysis

- As for Table 1, there are no totals in the different variables, and in some of them there are missing data and in others not. And, more important, the totals do no add to 120 in some of the cases (ie. Turn shift, age, training..). The percentages are not well calculatred in other cases (i,e. Ongoing training…)

- There are not meaningful descriptive results for the GHQ, Burnout, engagement and Job satisfaction questionaires, which makes it difficult to understand the conslussions

- The authors mention in the Discussion the influence of gender, but they do not show any results regarding this variable.

6. PLOS authors have the option to publish the peer review history of their article (what does this mean?). If published, this will include your full peer review and any attached files.

Reviewer #1: **Yes: **María Cantero-García

Reviewer #2: No

---

## [Author Response · Author response to Decision Letter 0]

13 Oct 2024

Dear Editor, 

We are very grateful to you for considering our article for possible publication in the Plos One Journal. 

We have received and studied the reviewers' comments. Thanks to your input, the study has been significantly improved. At the end of this letter, we present the comments received and how they have been addressed in the article. 

Regards. 

Paloma López Ros.

First, we have ensured that the manuscript complies with PLOS ONE style requirements, including those for file naming.

In addition, we have included the tables as part of the main manuscript and removed the individual files. 

We then proceed to answer the reviewers' questions.

REVIEWER 1

Selección de la muestra: The paper should provide a more detailed account of how the sample was selected. Specifically, could you elaborate on the criteria and process used for sample selection?

Thank you very much for your comment. The aspects indicated in the procedure section of the method section of the manuscript have been specified. The recruitment process has been explained in detail, it has been pointed out that the selection of 50% of the staff is random and the actual response rate has been added, indicating that it is not 100%.

The data analysis section mentions the use of non-parametric tests due to violations of normality and homogeneity assumptions. Could you provide more details on how these assumptions were tested before deciding to use non-parametric methods? Additionally, how were effect sizes interpreted, and what criteria were used to determine their relevance within the context of the study?

We welcome your input. A brief clarification has been added in the manuscript to answer your questions in the data analysis section. As mentioned, first, the Saphiro-Wilk test and Levene's test were used to analyse normality and homogeneity, respectively. When Levene's test is significant (p > .05), the assumption of homogeneity of variances is violated. At the same time, a low p-value in the Saphiro-Wilk test implies a breach of the normality assumption. This is what happened with the data analysed in our sample. Moreover, being a relatively small sample, we opted for the use of non-parametric statistics. This is also advised by the evidence [1].

Regarding the second question, the effect size was estimated with the epsilon squared statistic (ε2), considering for its interpretation Cohen's criteria for the partial eta2 analogue test: 0.01 (small), 0.06 (moderate) and 0.14 (large). This is recommended by previous literature [2,3,4].

Results Discussion: The results should be discussed in greater depth. Particularly, it would be helpful to better justify why administrative staff seem to be more exposed to non-physical violence. How do these findings compare with existing literature on this topic?

We welcome your suggestion. We have added in the second paragraph of the discussion information on the causes of increased exposure to user violence by administrative staff.

Lateral Violence: Regarding lateral violence, how do your findings align with the existing literature on this type of violence in healthcare settings? What specific factors might be contributing to the observed differences in prevalence among the various professional groups?

Thank you very much for your input, we think this is a very good point to strengthen the discussion section. It has been added what other studies have found regarding lateral violence in healthcare settings. Also, information has been added on why there are differences between countries on the professional group

Practical Recommendations: While the discussion covers the implications of the results well, it would benefit from a more detailed exploration of practical recommendations for managing workplace violence based on your findings. What specific intervention programs would you propose for administrative staff and healthcare professionals?

Thank you very much for your comment, we believe it can contribute a lot to our work. At the end of the discussion section we have added information on what these programmes could be about according to the literature.

Future Research: The discussion would benefit from a more comprehensive reflection on future research directions. What additional studies do you suggest to further explore the issues identified in your research?

Thank you for your comment. We have elaborated at the end of the discussion section on future research that could be carried out on the basis of the results of our work.

REVIEWER 2

The authors do not specify clearly the procedure for the recruitement of the participants. They select 50 % of the staff of the Emergency departments involved but it is not clear if they are randomly selected or not. It seems that the response rate is 100 %, which seems rare in a questionaire with 116 items.

Thank you very much for your comment. The aspects indicated in the procedure section of the method section of the manuscript have been specified. The recruitment process has been explained in detail, it has been pointed out that the selection of 50% of the staff is random and the actual response rate has been added, indicating that it is not 100%.

- There is too much confussion regarding the profesional profile of the participants: sometimes the authors refer to Non-medical staff and sometimes to Administrative assistants or Adminsitrators. 

Thank you for your comments. We have revised the entire manuscript, changing this term.

And the “Other professionals” category sometimes is included in the analysis and sometimes not.

We greatly appreciate your appreciation as this was a mistake on our part and greatly improves our study. In the first part of the results (Table 2), the results of the category "Other professionals" have been added. Again, thank you very much for your review.

The authors do not specify how they manage the results of the questionaires to analyze them: It seems that they use a collapsed measure of the different factors, but they not specify how these measures are constructed from the original ítems. 

Thank you very much for your comment. The information requested in the manuscript has been added. Scores are obtained from the mean of the set of items belonging to each variable. Specifically:

- Non-physical user violence (mean of items 18-24). 

- Physical user violence (mean of items 25-27). 

- Lateral personal violence (mean of items 28, 29, 30, 33). 

- Lateral relational violence (mean of items 31, 32, 34). 

- Lateral work-related violence (mean of items 35-37). 

- Depression (mean of items 66-72) 

- Anxiety (mean of items 52-58) 

- Somatic symptoms (mean of items 45-51) 

- Social dysfunction (mean of items 59-65) 

- Emotional exhaustion (mean of items 73, 74, 75, 76 and 78)

- Professional efficacy (mean of items 77, 79, 82, 83, 84 and 87) 

- Cynicism (mean of items 84 and 87), 84 and 87) 

- Cynicism (mean of items 80, 81, 85, 86) 

- Vigor (mean of items 88, 89, 92) 

- Dedication (mean of items 90, 91, 94) 

- Absorption (mean of items 93, 95, 96) 

- Intrinsic job satisfaction (mean of items 193, 194, 195, 196, 197, 200, 202, 204, 205, 206, 208, 212). 

- Extrinsic Job Satisfaction (mean of items 198, 199, 201, 203, 210, 211) 

- Overall Job Satisfaction (mean of items 207, 209)

Besides, they use in Table 2 a YES/NO category for user and lateral violences, but in Table 3 they use a seemingly Likert scale for the same variable

We appreciate your comment, this point has been specified in the manuscript. In addition, we would like to clarify the confusion regarding Table 2. The questionnaires assessing user and interprofessional violence, HABS-U and HABS-CS, respectively, are Likert-type, with 6 response options ranging from 1="Never" to 6="Daily". The question asked in both questionnaires is: "How often are you exposed to...". And the items are different situations of violence. It is understood that when a professional marks an item with "2=Annually", "3=Quarterly", "4=Monthly", "5=Weekly" or "6=Daily"; that subject is considered to be in the "YES" category. That is to say, at some time in the last year he/she has been exposed to some situation of violence at work. In this way, the percentages of perceived violence in the workplace by occupational group and year were calculated.

There is no bibliographical reference for the JAMOVI program used in the analysis

Thank you very much for your suggestion. The corresponding reference has been added. We regret the error and greatly appreciate your help in improving the work.

As for Table 1, there are no totals in the different variables, and in some of them there are missing data and in others not. And, more important, the totals do no add to 120 in some of the cases (ie. Turn shift, age, training..). The percentages are not well calculatred in other cases (i,e. Ongoing training…)

We thank you for your observation and deeply regret the missing data, this was an error on our part and your review helps us to improve our study. The data in Table 1 have been thoroughly reviewed, total scores and missing data have been added for those variables where they were missing. In addition, the percentages have been reviewed to ensure that they were correctly calculated. Again, many thanks for your comments.

There are not meaningful descriptive results for the GHQ, Burnout, engagement and Job satisfaction questionaires, which makes it difficult to understand the conslussions

Thank you very much for your comment, it is true that our study does not show significant results on these variables. This is contrary to what is expected based on previous literature in other populations or other related studies that point in the direction that there could be differences. Therefore, we consider it important to open such a debate in the discussion. The non-significance of our results may be due to the characteristics of our sample.... Therefore, future lines of research are proposed to further investigate this issue. This idea has been emphasized in the manuscript for clarification, thank you very much.

The authors mention in the Discussion the influence of gender, but they do not show any results regarding this variable.

We appreciate your comment. It is true that we did not measure the influence of gender in our results. However, it is an interesting debate to discuss, along the lines of future research proposals in relation to our study. The discussion of an article goes beyond the results obtained in it, the idea is to deepen and discuss what is known so far on the subject and in what line it would be interesting to continue researching. Once again, we thank you for your comments.

REFERENCIAS:

1-Ato García M, Vallejo Seco G. Diseños de investigación en psicología. Madrid: Pirámide; 2015.

2-Cohen J.Statistical power Analysis Jbr the Behavioral Sciences. Hillsdale (NJ): Lawrence Erlbaum Associates; 1988.

3-Bahcivan, Ozan; Estapé, Tania; Gutierrez-Maldonado, Jose. Efficacy of new mindfulness-based swinging technique: a pilot randomised controlled trial among women with breast cancer. Front. Psychol. 2022 Jul; 13:863857. 

4-Vargas Rubilar, Natalia; Oros, Laura Beatriz. Stress and Burnout in Teachers During Times of Pandemic. Front. Psychol. 2021 Nov; 12:756007.

---

## [Decision Letter · Decision Letter 1]

21 Oct 2024

PONE-D-24-23201R1DIFFERENCES IN WORKPLACE VIOLENCE AND HEALTH VARIABLES AMONG PROFESSIONALS IN A HOSPITAL EMERGENCY DEPARTMENT: A DESCRIPTIVE-COMPARATIVE STUDYPLOS ONE

Dear Dr. López-Ros,

Thank you for submitting your manuscript to PLOS ONE. After careful consideration, we feel that it has merit but does not fully meet PLOS ONE’s publication criteria as it currently stands. Therefore, we invite you to submit a revised version of the manuscript that addresses the points raised during the review process.

 Please see the comments below. 

We look forward to receiving your revised manuscript.

Kind regards,

Alejandro Botero Carvajal, MD

Academic Editor

PLOS ONE

Reviewers' comments:

Reviewer's Responses to Questions

**Comments to the Author**

1. If the authors have adequately addressed your comments raised in a previous round of review and you feel that this manuscript is now acceptable for publication, you may indicate that here to bypass the “Comments to the Author” section, enter your conflict of interest statement in the “Confidential to Editor” section, and submit your "Accept" recommendation.

Reviewer #1: All comments have been addressed

Reviewer #2: (No Response)

2. Is the manuscript technically sound, and do the data support the conclusions?

Reviewer #1: Yes

Reviewer #2: Partly

3. Has the statistical analysis been performed appropriately and rigorously? 

Reviewer #1: Yes

Reviewer #2: N/A

4. Have the authors made all data underlying the findings in their manuscript fully available?

Reviewer #1: Yes

Reviewer #2: No

5. Is the manuscript presented in an intelligible fashion and written in standard English?

Reviewer #1: Yes

Reviewer #2: Yes

6. Review Comments to the Author

Reviewer #1: The authors have made all the changes excellently, addressing all feedback thoroughly. Great work overall!

Reviewer #2: There are still problems with the data presented in table 1 (to be revised again by the authors).

In page 4 the authors state that the study population consists of 120 professionals, but it seems that this number is the final number of professionals that answered the questionarires (so, the size of the final sample).

There continues to be problems in the way the authors describe the sample selection (at least they should specify the total size of the population, the selection method, the response rate, and some comments and data on the differences between the initial sample and the final sample).

There continues to be some problems with the categorization of the professionals. And some curious result: It seems that physicians show higher levels of satisfaction than medical staff (which is the differene between physicians and medical staff?): this should be adressed ensuring that the names of the different categories of study are the same through all the manuscript, including the tables.

Regarding the gender issue: as the authors have data on the gender of the professionals, if they think that gender is a relevant issue, they should present the data and conclussions of this study relative to gender. The discussion section must be generated by the comparison of the data and conclussions of the study and other relevant comparable studies in the bibliography.

The same applies to the discussion on job satisfaction, burnout or engagement: The authors make comments on different aspects that influence these variables, but they do not compare with the data on their study, making the discussion less meaningful.

7. PLOS authors have the option to publish the peer review history of their article (what does this mean?). If published, this will include your full peer review and any attached files.

Reviewer #1: **Yes: **María Cantero-García

Reviewer #2: No

---

## [Author Response · Author response to Decision Letter 1]

23 Oct 2024

Dear Editor, 

We are very grateful to you for considering our article for possible publication in the Plos One Journal. 

We have received and studied the reviewers' comments. Thanks to your input, the study has been significantly improved. At the end of this letter, we present the comments received and how they have been addressed in the article. 

Regards. 

Paloma López Ros.

We then proceed to answer the reviewers' questions.

REVIEWER 2

There are still problems with the data presented in table 1 (to be revised again by the authors).

We are grateful for your comment. We are sorry for the mistakes in this regard. The relevant errors have been rechecked and amended. Again, we thank you for your observations that have improved our study.

In page 4 the authors state that the study population consists of 120 professionals, but it seems that this number is the final number of professionals that answered the questionarires (so, the size of the final sample).

Thanks for your comment. It has been specified that, indeed, the final sample size is 120.

There continues to be problems in the way the authors describe the sample selection (at least they should specify the total size of the population, the selection method, the response rate, and some comments and data on the differences between the initial sample and the final sample).

Thanks for your comment. This information is described in detail in the Procedure section

There continues to be some problems with the categorization of the professionals. And some curious result: It seems that physicians show higher levels of satisfaction than medical staff (which is the differene between physicians and medical staff?): this should be adressed ensuring that the names of the different categories of study are the same through all the manuscript, including the tables..

Thanks for your feedback. Your comments highly improve our study. We have made sure that the different categories have the same name throughout the study so that there is no confusion.

Regarding the gender issue: as the authors have data on the gender of the professionals, if they think that gender is a relevant issue, they should present the data and conclussions of this study relative to gender. The discussion section must be generated by the comparison of the data and conclussions of the study and other relevant comparable studies in the bibliography.

The same applies to the discussion on job satisfaction, burnout or engagement: The authors make comments on different aspects that influence these variables, but they do not compare with the data on their study, making the discussion less meaningful..

We greatly appreciate your comment. We believe that the discussion includes a comparison of our results and conclusions with those of previous literature. This is reflected, for example, in the second and third paragraphs, where we specifically discuss the differences found between practitioners. In the sixth, seventh and eighth paragraphs, you can also find a comparison with previous evidence. This covers the main objective of our study, to analyze the differences between the different professional categories. 

It is a very interesting proposal to analyze gender and other variables in relation to workplace violence, satisfaction, burnout, engagement... However, this differs from the aim of our study and would unfortunately exceed the length of the manuscript. 

In our discussion we have considered encompassing a broader vision that contextualizes the reader on all the influential variables according to the subject matter, going beyond the results obtained in the study. We consider this to be an interesting debate to discuss, in line with future research proposals in relation to our study, as our main objective. Once again, thank you very much for your comment.

---

## [Decision Letter · Decision Letter 2]

30 Oct 2024

PONE-D-24-23201R2DIFFERENCES IN WORKPLACE VIOLENCE AND HEALTH VARIABLES AMONG PROFESSIONALS IN A HOSPITAL EMERGENCY DEPARTMENT: A DESCRIPTIVE-COMPARATIVE STUDYPLOS ONE

Dear Dr. López-Ros,

Thank you for submitting your manuscript to PLOS ONE. After careful consideration, we feel that it has merit but does not fully meet PLOS ONE’s publication criteria as it currently stands. Therefore, we invite you to submit a revised version of the manuscript that addresses the points raised during the review process.

We look forward to receiving your revised manuscript.

Kind regards,

Alejandro Botero Carvajal, MD

Academic Editor

PLOS ONE

Journal Requirements:

Reviewers' comments:

Reviewer's Responses to Questions

**Comments to the Author**

1. If the authors have adequately addressed your comments raised in a previous round of review and you feel that this manuscript is now acceptable for publication, you may indicate that here to bypass the “Comments to the Author” section, enter your conflict of interest statement in the “Confidential to Editor” section, and submit your "Accept" recommendation.

Reviewer #2: (No Response)

2. Is the manuscript technically sound, and do the data support the conclusions?

Reviewer #2: Yes

3. Has the statistical analysis been performed appropriately and rigorously? 

Reviewer #2: Yes

4. Have the authors made all data underlying the findings in their manuscript fully available?

Reviewer #2: Yes

5. Is the manuscript presented in an intelligible fashion and written in standard English?

Reviewer #2: Yes

6. Review Comments to the Author

Reviewer #2: This paragraph in the Results section of the Summary should be revised as it is incongruent (physicians cannot show hiogher levels of satisfaction than medical staff as they are the same group):

"Regarding job satisfaction, nurses report higher intrinsic satisfaction, while physicians, nurses and auxiliary nursing staff show higher levels of extrinsic satisfaction compared to administrative and medical staff."

In the same direction, this paragraph in the Results section of the text should be revised (lines 324-326) as it is incongruent (medical staff cannot at the same time show greater and lower extrinsic satisfaction thanm administration group):

"In extrinsic satisfaction, medical staff, nursing staff and nursing assistants showed significantly higher scores than the administration group and this, in turn, than the medical staff"

7. PLOS authors have the option to publish the peer review history of their article (what does this mean?). If published, this will include your full peer review and any attached files.

Reviewer #2: No

---

## [Author Response · Author response to Decision Letter 2]

1 Nov 2024

Dear Editor, 

We are very grateful to you for considering our article for possible publication in the Plos One Journal. 

We have received and studied the reviewers' comments. Thanks to your input, the study has been significantly improved. At the end of this letter, we present the comments received and how they have been addressed in the article. 

Regards. 

Paloma López Ros.

We then proceed to answer the reviewers' questions.

REVIEWER 2

This paragraph in the Results section of the Summary should be revised as it is incongruent (physicians cannot show hiogher levels of satisfaction than medical staff as they are the same group):

"Regarding job satisfaction, nurses report higher intrinsic satisfaction, while physicians, nurses and auxiliary nursing staff show higher levels of extrinsic satisfaction compared to administrative and medical staff."

We are grateful for your comment. We are sorry for the mistakes in this regard. The relevant errors have been rechecked and amended. 

“Regarding job satisfaction, nurses report higher intrinsic satisfaction. Medical staff, nurses and nursing assistants show higher levels of extrinsic satisfaction compared to administrative staff”

In the same direction, this paragraph in the Results section of the text should be revised (lines 324-326) as it is incongruent (medical staff cannot at the same time show greater and lower extrinsic satisfaction thanm administration group):

"In extrinsic satisfaction, medical staff, nursing staff and nursing assistants showed significantly higher scores than the administration group and this, in turn, than the medical staff"

Thank you for your comment. We have clarified this section and corrected it in the manuscript.

“In extrinsic satisfaction, medical staff, nursing staff and nursing assistants showed significantly higher scores than the administration group (χ²= 30.21; p< .001). In this case, the magnitude of the effect was high “

---

## [Decision Letter · Decision Letter 3]

19 Nov 2024

DIFFERENCES IN WORKPLACE VIOLENCE AND HEALTH VARIABLES AMONG PROFESSIONALS IN A HOSPITAL EMERGENCY DEPARTMENT: A DESCRIPTIVE-COMPARATIVE STUDY

PONE-D-24-23201R3

Dear Dr. López-Ros,

We’re pleased to inform you that your manuscript has been judged scientifically suitable for publication and will be formally accepted for publication once it meets all outstanding technical requirements.

Kind regards,

Alejandro Botero Carvajal, MD

Academic Editor

PLOS ONE

Additional Editor Comments (optional):

Reviewers' comments:

Reviewer's Responses to Questions

**Comments to the Author**

1. If the authors have adequately addressed your comments raised in a previous round of review and you feel that this manuscript is now acceptable for publication, you may indicate that here to bypass the “Comments to the Author” section, enter your conflict of interest statement in the “Confidential to Editor” section, and submit your "Accept" recommendation.

Reviewer #2: All comments have been addressed

2. Is the manuscript technically sound, and do the data support the conclusions?

Reviewer #2: Yes

3. Has the statistical analysis been performed appropriately and rigorously? 

Reviewer #2: Yes

4. Have the authors made all data underlying the findings in their manuscript fully available?

Reviewer #2: Yes

5. Is the manuscript presented in an intelligible fashion and written in standard English?

Reviewer #2: Yes

6. Review Comments to the Author

Reviewer #2: (No Response)

7. PLOS authors have the option to publish the peer review history of their article (what does this mean?). If published, this will include your full peer review and any attached files.

Reviewer #2: No

---

## [Editor Report · Acceptance letter]

25 Nov 2024

PONE-D-24-23201R3 

PLOS ONE

Dear Dr. López-Ros, 

I'm pleased to inform you that your manuscript has been deemed suitable for publication in PLOS ONE. Congratulations! Your manuscript is now being handed over to our production team.

Kind regards, 

on behalf of

Dr. Alejandro Botero Carvajal 

Academic Editor

PLOS ONE